# Detecting Cortical Thickness Changes in Epileptogenic Lesions Using Machine Learning

**DOI:** 10.3390/brainsci13030487

**Published:** 2023-03-14

**Authors:** Sumayya Azzony, Kawthar Moria, Jamaan Alghamdi

**Affiliations:** 1Department of Computer Sciences, Faculty of Computing and Information Technology, King Abdulaziz University, Jeddah 21589, Saudi Arabia; 2Diagnostic Radiology Department, Faculty of Applied Medical Sciences, King Abdulaziz University, Jeddah 21589, Saudi Arabia

**Keywords:** epilepsy, drug-resistant epilepsy, magnetic resonance imaging, cortical thickness, cerebrospinal fluid, machine learning

## Abstract

Epilepsy is a neurological disorder characterized by abnormal brain activity. Epileptic patients suffer from unpredictable seizures, which may cause a loss of awareness. Seizures are considered drug resistant if treatment does not affect success. This leads practitioners to calculate the cortical thickness to measure the distance between the brain’s white and grey matter surfaces at various locations to perform a surgical intervention. In this study, we introduce using machine learning as an approach to classify extracted measurements from T1-weighted magnetic resonance imaging. Data were collected from the epilepsy unit at King Abdulaziz University Hospital. We applied two trials to classify the extracted measurements from T1-weighted MRI for drug-resistant epilepsy and healthy control subjects. The preprocessing sequence on T1-weighted MRI images was performed using C++ through BrainSuite’s pipeline. The first trial was performed on seven different combinations of four commonly selected measurements. The best performance was achieved in Exp6 and Exp7, with 80.00% accuracy, 83.00% recall score, and 83.88% precision. It is noticeable that grey matter volume and white matter volume measurements are more significant than the cortical thickness measurement. The second trial applied four different machine learning classifiers after applying 10-fold cross-validation and principal component analysis on all extracted measurements as in the first trial based on the mentioned previous works. The K-nearest neighbours model outperformed the other machine learning classifiers with 97.11% accuracy, 75.00% recall score, and 75.00% precision.

## 1. Introduction

Epilepsy is a neurological disorder. It is characterized by abnormal brain activity. Epileptic patients suffer from unpredictable seizures which cause changes in their behaviour, movement, feelings, and sometimes causes a loss of awareness. The World Health Organization (WHO) estimates that epilepsy affects over 50 million people globally; this large number makes epilepsy the most common neurological disorder. Around three-quarters of epilepsy patients in low- and middle-income countries do not receive the treatment they require, and they and their families face stigma and discrimination in many parts of the world, as per the WHO’s report. Approximately 30% of those patients suffer from refractory seizures [1]. Seizures are considered refractory or drug-resistant when treatment does not succeed in achieving seizure freedom for at least 12 months for any reason [2] or if the treatment fails two or more medicines and a seizure occurs once or more per month over 18 months [3]. During the presurgical workup of drug-resistant epilepsy, specialists seek to determine focal epileptogenic brain lesions to apply resection for those lesions which are predicted to cure epilepsy after resection. Focal cortical dysplasia (FCD) is an epileptogenic lesion that requires surgical intervention for treatment of epilepsy. It is a type of cortical development malformation distinguished by disturbed cortical lamination, balloon cells, dysmorphic neurons, and/or ectopic neurons in the white matter [4,5,6]. FCD type I to FCD type III encompasses a wide range of histological and clinical features [4]; although certain FCDs are easily detectable with traditional neuroimaging, others are more subtle [7]. Neuroradiologists have reported detection rates of FCD type II lesions ranging from 65 to 91 percent [8,9]. Figure 1 shows an example of an FCD lesion. Mesial temporal sclerosis (MTS) is another typical epileptogenic lesion in drug-resistant epilepsy that requires surgical intervention. Figure 2 shows an MTS lesion.

Specialists use multiple sources of data to detect these lesions, such as electron-encephalographic (EEG), positron emission tomography (PET), brain magnetic resonance imaging (MRI), and others. Although this advanced imaging is used for detection of these subtle morphological abnormalities in epilepsy patients, the diagnosis of FCD and MTS requires visual inspection by expert neuroradiologists. Recent advancements in MRI have made it easier to detect and diagnose FCD and other cortical malformations (CM) in epileptic patients [12].

Because the human body is made up of molecules including nuclei, MRI scanners employ the electromagnetic activity of these nuclei along with solid magnetic fields and radio wave to build images of the body. The hydrogen atom is often employed in MRI investigations because a major fraction of the human body consists of fat and water, both of which comprise many hydrogen atoms [13]. In MRI-negative epilepsy, FCD is a widespread pathology [14]. Quantitative post-processing methods, such as voxel-based morphometric analysis and surface-based feature analysis, have been developed to address some of the limitations of visual detection of FCD with MRI. Voxel-based morphometry algorithm [9,15,16,17], surface-based morphometry algorithm [13,14], and the post-processing method [15] are common approaches for detecting epileptic foci. The image density is compared to a normal template in the voxel-based morphometry approach, and the area that is abnormally identified in the image is considered as the lesion area. The surface-based morphometry method is primarily applied to rebuild the cerebral cortex, extract useful features, and classify and locate the lesion area using a machine learning method. The post-processing procedure is utilized to extract parameters such as texture and cortical thickness, build a computational model, and pinpoint the location of the lesion. Currently, two new works have used an advanced convolutional neural network to investigate FCD lesion detection, and they have yielded promising findings [18,19]. The cerebral cortex in humans, a narrow ribbon of grey matter that makes up the cerebrum’s outer layer, is around 2.5 mm thick on average [20]. Cortical thickness is the length of the distance between the white and grey matter surfaces of the brain at various locations [20,21]. It is also sensitive to variations in health, such as ageing and diseases such as schizophrenia, Alzheimer’s, and depression [22,23,24]. It has been shown in numerous investigations to be a surrogate marker for underlying disease abnormalities [25,26,27]. With normal ageing, cortical thickness diminishes in Alzheimer’s disease (AD) [28]. Furthermore, it increases in temporal lobe epilepsy (TLE).

An important component of the field of artificial intelligence is machine learning (ML). Without requiring any explicit programming, ML algorithms create a model from training data to produce predictions or judgments [29]. Various ML algorithms are used to classify diseases. Logistic regression (LR), k-nearest neighbours (K-NN), support vector machine (SVM), and naive Bayes are common ML algorithms. This research examines these four classifiers to build a high-performance model of T1-weighted MRI images in drug-resistant epilepsy classification.

### 1.1. Logistic Regression (LR)

This statistical analysis method is frequently used for applications in predictive analytics, modelling, and machine learning. The dependent variable in this analytics method may be either categorical or numerical. Any of four finite possibilities, A, B, C, or D, or in the case of binary regression, either A or B, are possible outcomes (multiple regression). Statistical software uses a logistic regression equation to compute probabilities and determine the relationship between the dependent variable and one or more independent variables [30].
logistic(η)=1/(1+exp(−η))

### 1.2. K-Nearest Neighbours (K-NN)

The non-parametric supervised learning method known as the k-nearest neighbours algorithm (k-NN) was created in 1951 [31] and subsequently developed in statistics [32]. It is employed in the categorization and regression of data. The input in both situations consists of the k closest training examples in a data collection. The outcome of k-NN classification is a class membership. A majority vote of its neighbours categorizes an object, and each object is allocated to the most common class among its k closest neighbours (k is a positive integer, typically small). If k = 1, the object is assigned to the nearest neighbour’s class.

### 1.3. Support Vector Machine (SVM)

SVM is one of the most extensively used machine learning methods in medical image processing. It is a supervised algorithm first introduced by Vishwanathan and Murty [33], and it has been developed over the years. It performs classification and regression analysis on data. SVM divides the data into classes in the training sets by identifying the best decision boundary. This aids in classifying new data added to the correct side of the separator, known as a hyperplane. The hyperplane is generated by selecting a data set’s extreme points, which are referred to as support vectors. Figure 3 shows the SVM hyperplane for linearly separable data points.

### 1.4. Naive Bayes

Simple probabilistic classifiers known as naive Bayes classifiers are based on the Bayes theorem and strong (naive) independence assumptions between statistical features. Although they are among the simplest Bayesian network models, they can attain high levels of accuracy when paired with kernel density estimation [35]. The number of variables (features) in a learning issue is linear in the number of parameters used by naive Bayes classifiers. Maximum-likelihood training can be carried out by simply evaluating a closed-form expression in linear time instead of the costly iterative approximation employed for many other types of classifiers [36].

## 2. Literature Review

In previous studies, machine learning (ML) techniques have shown consistently strong discriminatory capacity across the broad variety of epilepsy applications [37]. Furthermore, deep learning (DL) has recently been applied in various fields in epilepsy. Morphometric analysis program (MAP) strategies have been performed in this field of detection. We focused on analysis methods to categorize the previous studies, as you will read below.

### 2.1. Morphometric Analysis Program (MAP) Strategies for FCD Detection

MAP was first launched in 2005 [16], was independently evaluated for clinical advantages, and has since been effectively integrated into regular presurgical workflows at a number of epilepsy facilities around the world [9,38]. The diagnostic utility of MAP in focal drug-resistant epilepsy with FCD is investigated in [17]. In this work, for each of the 39 individuals with FCD, an automated MAP created z-score maps resulting from T1-weighted MRI scans, which were compared to healthy adults or healthy pediatricians. Independent of various imaging modalities and clinical data, MAP detected abnormal grey matter extension into white matter and blurring of the grey–white matter junction. The specificity and sensitivity of the extension and junction MAP were higher than the qualitative MRI. T. Demerath et al. [39] performed a comparison between MP2RAGE and MPRAGE, the most common sequence for 3D T1-weighted imaging in Siemen’s scanners, in FCD detection. A total of 640 epilepsy patients were studied, and the results indicated FCD lesions were clearly detected in MP2RAGE junction images, whereas two were not seen in MPRAGE junction images. Both FCD volume and z-scores of mean lesions were larger in the MP2RAGE junction images than in the MPRAGE-based images. Based on the MAP18 morphometric output maps, Bastian David et al. [40] constructed a feed-forward artificial neural network to detect FCD. The artificial neural network was cross-validated and trained on a manually separated data set comprising 113 patients with FCD and 362 healthy controls. On 60 FCD samples, 70 healthy controls, and an unseen data set, they confirmed the trained artificial neural network’s performance. On the training data set, the artificial neural network had 87.4% sensitivity and 85.4% specificity in cross-validation. Their technique still had a sensitivity of 81.0% on the separated validation data set, with a similar high specificity of 84.3%.

### 2.2. Temporal Lobe Epilepsy Applications Based on Machine Learning (ML)

In [41], an SVM technique was used to predict temporal lobe epilepsy based on mean kurtosis, mean diffusivity, and fractional anisotropy from three separate imaging modalities. Whereas classic imaging of diffusion tensor can be used to calculate mean diffusivity and fractional anisotropy, diffusion kurtosis imaging is required to calculate mean kurtosis. Diffusion kurtosis imaging was used on 32 temporal lobe epilepsy cases and 36 healthy cases. A 1000-iteration five-fold cross-validation was used to measure prediction capability. The topic photos in the training set were used to train SVM models, each with a different regularization parameter, and their performances were measured on the test set. A Bayesian-based technique was used to identify the various regularization values. On every iteration, mean kurtosis outperformed fractional anisotropy and mean diffusivity and had considerably greater average accuracy, with 82% mean kurtosis, 68% fractional anisotropy, and 51% mean diffusivity. Esmaeil Davoodi-Bojd et al. [42] classified temporal lobe epilepsy lateralization using linear SVM. To determine the most efficient connections for lateralizing the disease, they examined the connectivity matrices produced from diffusion-weighted MRI of 10 left and 10 right patients. Their results showed an accuracy of 100%, although that high accuracy may be due to the small population. For temporal lobe detection, Kouhei Kamiya et al. [43] applied supervised machine learning via SVM of diffusion tensor imaging structured brain connectomes of forty-four patients: fifteen right, twenty-nine left, and fourteen age-matched controls. Their method showed 75.9–89.7% accuracy for the right temporal lobe epilepsy versus controls, 74.4–86.0% accuracy for the left temporal lobe epilepsy versus controls, and 72.7–86.4% accuracy for the left temporal lobe epilepsy versus right temporal lobe epilepsy. Seventeen anatomical MRI data of normal and epileptic cases were used for training SVM classifiers to diagnose epilepsy. They compared the two different methods for MRI segmentation: the unified segmentation method, and the Gram–Schmidt orthogonalization method. They concluded that the unified segmentation method outperforms the Gram–Schmidt orthogonalization method. The best accuracy they obtained using the whole-brain analysis approach and the unified segmentation method was 94% [44].

#### 2.2.1. Automated Detection of FCD Using ML

ML methods can provide a major FCD diagnosis outcome as a presurgical assessment for drug-resistant epilepsy patients. Zohera et al. [45] applied artificial neural networks after extracting morphological and intensity-based features. They used a total of 58 patients: 30 with verified FCD type II and 28 adults as healthy controls. The classification accuracy levels for lobe and hemisphere detection in the region where the lesion was found were 84.2 and 77.3, respectively. In [46], the authors introduced a useful method for detecting non-temporal lobe lesions. They detected FCD lesions using negative images of FLAIR based on cortical thickness features calculated using the Laplace method. They obtained cortical thickness average images and standard deviations for 32 healthy control subjects. After that, they subtracted the cortical thickness average images from the images of cortical thickness of each patient and divided the output by the cortical thickness standard deviation image to compute an extension map of cortical thickness. Lastly, they found that a cluster with greater than three voxels was considered to be an FCD lesion area. Ravnoor S. Gill et al. [47] proposed a novel algorithm that combined surface-based morphometry and intensity features to detect FCD using multi-modal MRI. They evaluated their approach using five-fold cross-validation and obtained values of 83% sensitivity and 92% specificity. Meriem El Azami et al. [48] proposed a computer-aided diagnosis system to detect lesions underlying drug-resistant epilepsy in T1-weighted MRI. They used two features: heterotopia, where tissue follows grey matter for all sequences and the margin often is not clear, and the presence of a blurred junction between the white matter and the grey matter. Seventy-seven healthy controls and eleven patients with thirteen lesions were used. A one-class support vector machine was used to classify lesions. Their proposed method detected all lesions successfully.

In [49], research was conducted on 61 patients with drug-resistant epilepsy. Three distinct MRI scanners were used to analyse the patients at three different epilepsy centres. With 120 healthy controls, a normal database was created. To determine specificity, they used thirty-five healthy controls and fifteen illness controls with hippocampal sclerosis for testing. A non-linear neural network was trained to identify lesioned clusters using features that were calculated and included in it. They used analysis of receiver operating characteristic to improve the threshold of the probability map classifier’s output. The intersection between the manual labelling and final cluster was used to determine detection success. K-fold cross-validation was used to assess performance. The 0.9 thresholds resulted in 73.7% sensitivity and 90.0% specificity. The analysis of ROC had an area under the curve of 0.75. Bilal Ahmed et al. [50] applied an automated quantitative morphometry approach to compute surface-based MRI features and combine them in a model for classifying lesional and non-lesional vertices. Their model detected 6 out of 7 MRI-positive patients correctly and 14 out of 24 MRI-negative patients, which is considered to be a good achievement in MRI-negative prediction. Sophie Adler et al. [51] added new surface-based measurements such as local cortical deformation and per-vertex interhemispheric asymmetry in addition to traditional features such as cortical thickness, blurring of grey matter, and many others to detect FCD in pediatric epilepsy. After inter- and intra-subject normalization using twenty-eight healthy controls, an artificial neural network classifier was built using data from twenty-two focal epilepsy patients. They obtained higher sensitivity (73%) values when using the novel measures (excluding established measures) than when using only the established measures (59%). Jia-Jie Mo et al. [52] combined an artificial neural network with quantitative multi-modal surface-based features from T1-MPRAGE, FLAIR, and PET to automatically detect FCD lesions. Morphological features, intensity features, and metabolic features were calculated to input into the artificial neural network. To diminish the dimensions of the features, principal component analysis was used. The sensitivity, accuracy, and specificity of their ANN classifier were 70%, 70.5%, and 69.9% respectively.

#### 2.2.2. Automated Detection of MTS Using ML

The primary intention of this study [53] was to determine if functional connectivity in the resting state of magnetoencephalography (MEG) signals are able to be used as a biomarker for distinguishing mesial temporal lobe epilepsy patients from healthy patients as well as right and left mesial temporal lobe epilepsy patients. Among the different machine learning techniques used, an SVM method was used. They examined functional resting-state networks in 46 patients with mesial temporal lobe epilepsy (23 on the right; 23 on the left) who were seizure-free after surgery and 46 patients with hippocampal sclerosis. The best SVM group classifier identified mesial temporal lobe epilepsy patients with a 95.1% mean accuracy (95.8% sensitivity, and 94.3% specificity). Huiquan Wang et al. [11] proposed a method incorporating cerebrospinal fluid ratio features with hippocampal volume and shape features to ensure that the surrounding tissue properties of the hippocampus are included. They used T1-weighted MRI images, with fifteen normal controls, eighteen left MTS, and eighteen right MTS. Then, they used SVM to classify the type of MTS. Their proposed method resulted in 94% sensitivity, 100% specificity, and 97% accuracy. In a broad sample of epilepsy patients, ref. [54] created an automated ML methodology that was capable of identifying MTS with an accuracy of up to 81%. To create classification accuracy, the program took into account surface area, cortical thickness, and curvature of the surrounding temporal, frontal, and limbic structures in addition to hippocampus volume. MTS was linked to an younger age of onset, a longer period of disease, and more repeated seizures in people who were diagnosed.

### 2.3. Lesions Detection Based on Deep Learning (DL)

Qu, Guiguo and Yuan, Qi [55] introduced a new strategy for detecting epileptogenic regions based on convolutional neural networks and transfer learning to handle a small amount of data. They used the Bonn data set, which contains 100 focal and 100 non-focal EEG signals. The results of their experiments indicated that their technique was 95.00 accurate.

#### Automated Detection of FCD Using DL

Sixteen patients’ T1-weighted MRI images were separated into fourty-five structures. Each structure’s volume was compared at an individual level by matching gender and age to a normal population using a prototype made with MorphoBox, a software for brain volumetry. The prototype’s performance in patients was assessed using a receiver operating characteristics curve. Although their method could be used as a biomarker for detecting the lesions, it cannot stand alone for determining the specificity, location, and size of lesions. The result of their method showed 93.9% sensitivity, 79.6% specificity, and a receiver operating characteristics (ROC) curve of 0.89 [56]. In 2019, K.M. Bijay et al. [18] used a fully convolutional neural network-based model for the first time for detecting FCD lesions using only FLAIR images. Forty-three subjects were selected by a neurologist in a retrospective study. Their proposal method gave a patient-wise recall value of 82.5, region-wise values of 48 for recall and 89 for precision, and pixel-wise values of 40.1 for recall, 80.69 for precision, and 52.47 for the Dice coefficient. Ruslan Aliev et al. [57] automatically detected lesions of FCD using convolutional neural networks, and they proposed a new metric for the detection algorithm’s assessment. They applied their method on a data set with 15 labelled patients, and they obtained efficacious detection of FCD’s lesions in eleven out of fifteen subjects. Cuixia et al. [58] trained a six-layer convolutional neural network. Then, they performed activation maximization for identifying pattern image blocks that are highly similar to FCD lesion images using their network. They evaluated their technique with 19 negative lesion images from 12 FCD patients at an early time point to learn optimal cortical features automatically and enhance the detection of FCD. Their CNN architecture comprised five convolutional layers, two connected layers, and one pooling layer. Thirty T1-weighted MRI images were used. The patient group included ten healthy cases and ten temporal lobe epilepsy cases. Their experimental results showed a high performance of 90% sensitivity (in comparison to other state-of-the-art methods, which had a sensitivity of 70%), 85% specificity, and 88% accuracy. Based on literature review, we found that SVM is the most frequently used ML method in the literature.

## 3. Materials and Methods

This section details the sequence of the research methodology beginning with data acquisition, preprocessing, feature extraction, and classification. Figure 4 shows a schematic of the proposed T1-weighted MRI images in drug-resistant epilepsy classification. It includes four stages: data acquisition, preprocessing of data, feature extraction, and classification.

### 3.1. Data Acquisition

Eleven participants with drug-resistant epilepsy from the epilepsy unit at King Abdul-Aziz University Hospital were included in this study. In addition, eighteen healthy control subjects from the same hospital were included. The subjects are between 22 and 30 years old. Most of them are Saudi. A total of twenty-nine three-dimensional (3D) T1-weighted MRI images were provided, one for each subject. All images are in nifti format (.nii). Figure 5 shows an image from the collected data for the first patient.

### 3.2. Preprocessing

All preprocessing was performed using BrainSuite’s pipeline [59].

### 3.3. Feature Extraction

Eight measurements were calculated for 99 regions of interest (ROI) for a total of 792 features. The eight measurements are: mean thickness (mm), grey matter volume (mm${}^{3}$), cerebrospinal fluid volume (mm${}^{3}$), white matter volume (mm${}^{3}$), the total volume of grey matter and white matter (mm${}^{3}$), cortical area mid (mm${}^{2}$), cortical area inner (mm${}^{2}$), and cortical area pial (mm${}^{2}$). We excluded the features with values of zero, so we ended up with 664 features. Table 1 shows the names of the regions of interest.

Figure 6 shows some regions of interest in the brain.

### 3.4. ML Classifiers

This section explains the two trials that applied ML models. The first subsection describes the first trial, which applied SVM on different combinations of four selected measurements, which are: cortical thickness, grey matter volume, cerebrospinal fluid volume, and white matter volume. Furthermore, the second subsection describes the trial that applied four different ML classifiers to all measurements.

#### 3.4.1. First Trial on Four Commonly Selected Measurements

Based on our literature review, we found that SVM is the most frequently used ML classifier in drug-resistant epilepsy classification. Automated classification of four selected measurements was performed using an SVM classifier. The two classes that were used to classify the features were 0, which was used to label the healthy control subject, and 1, which was used to label the drug-resistant subject. Seven different combinations of four selected measurements (cortical thickness, white matter volume, grey matter volume, and cerebrospinal fluid volume) and a total of 355 features were used in this experiment. Different combinations were used to simplify the process. The combinations of measurements were used to choose the measurements that have the best effect on the classification process. Furthermore, in the literature, there was no clear evidence for selection of particular measurements. The combinations of measurements are denoted as follows:

Exp1: cortical thickness.

Exp2: white matter volume and grey matter volume.

Exp3: cerebrospinal fluid volume.

Exp4: white matter volume, grey matter volume, and cerebrospinal fluid volume.

Exp5: cortical thickness and cerebrospinal fluid volume.

Exp6: white matter volume, grey matter volume, and cortical thickness.

Exp7: cortical thickness, white matter volume, grey matter volume, and cerebrospinal fluid volume.

Feature scaling was applied to optimize the performance. A k-fold cross-validation (k = 10) strategy was used in each experiment to validate the classifier’s performance. Cross-validation is a statistical method for estimating machine learning models’ competence. It is a resampling technique for evaluating machine learning models on a small sample of data. K denotes the number of groups into which a given data sample will be divided. The following is the general procedure [61]:Shuffle the data set at random.Sort the data into k groups.For every separate group:(a)As a holdout or test data set, use the group.(b)As a training data set, use the remaining groups.(c)Fit a model to the training data and test it on the test data.(d)Keep the evaluation score but throw out the model.Provide a brief summary of the model’s skill using the sample of model evaluation scores.

This is based on the grid search strategy, which is an exhaustive search carried out on a model’s specific parameter values.

GridSearchCV [62] is a scikit-learn package in Python that allows tuning of the hyperparameters. The best parameters were selected for each experiment (see Table 2).

More details about the results will be provided in the next Section 4.2. The scikit-learn library version of support vector classification (SVC) may be configured with a large variety of hyperparameters, including:C: a parameter for regularization.Kernel: a parameter that can be set to linear, rbf, or our own callable.Gamma: the rbf, poly, and sigmoid kernel parameter coefficient.

#### 3.4.2. Second Trial on all Extracted Measurements

This subsection describes the classification of the extracted measurements using four different ML classifiers: logistic regression, k-nearest neighbours (K-NN), support vector machine (SVM), and naive Bayes.

Before applying the above-mentioned classifier, feature scaling was applied to optimize the performance as in the previous trial. In this trial, we used all eight measurements to include all calculated measurements and maximize the number of features. We eliminated the features with values of zero, leaving 664 features. Principal component analysis (PCA) was used to select the best features out of the 664.

For each classifier, a k-fold cross-validation (k = 10) strategy was used to validate the classifier’s performance.

## 4. Results and Discussion

This section explains the two trials’ results and performance.

### 4.1. Evaluation Metrics

In this subsection, descriptions and equations of the evaluation metrics are presented.

To evaluate a classifier’s performance, it is much better to examine the confusion matrix. The fundamental idea is based on counting the number of instances where representatives of class A are classified as class B [63]. The confusion matrix provides a lot of information. Still, we prefer a simpler statistic, such as accuracy, precision, and recall, which are described in the following three sections (Section 4.1.1, Section 4.1.2 and Section 4.1.3).

#### 4.1.1. Accuracy

The ratio of accurately predicted labels to the total number of predicted labels is calculated as follows. Equation (Equation 1). Accuracy
(1)$$Accuracy=\frac{TP+TN}{TP+FP+FN+TN}$$

The terms TP, TN, FP, and FN refer to the quantity of true positives, true negatives, false positives, and false negatives, respectively.

When it comes to classifiers, accuracy is not always the best metric to use, especially when working with skewed data sets (i.e., when some classes are much more frequent than others) [63] as in our case.

#### 4.1.2. Precision

The precision of the classifier, that is, the accuracy of positive predictions [63], is an interesting measurement to examine.

Equation (Equation 2). Precision
(2)$$Precision=\frac{TP}{TP+FP}$$

#### 4.1.3. Recall

Equation (Equation 3). Recall
(3)$$Recall=\frac{TP}{TP+FN}$$

### 4.2. Results

In this section, all the results of the two previously mentioned trials are listed and shown.

The first trial was an SVM classifier trained on different combinations of the four selected measurements: cortical thickness, grey matter volume, cerebrospinal fluid volume, and white matter volume. To evaluate the experiments, accuracy, recall score, and precision were calculated for each combination, as shown in Table 2. The Exp2, Exp6, and Exp7 combinations obtained the best accuracies, with 80.00%. GridSearchCV was used to select the best models’ parameters. Figure 7 shows accuracy, recall score, and precision of different combinations of measurements in this trial.

The second trial contains more intensive experiments. Four different ML classifiers were used on all extracted measurements, as shown in Table 3. Before training the classifiers, the principal component analysis (PCA) technique was applied to reduce the number of features. We found that the best number of features (n_components) was 20. Figure 8 shows the variance ratio classification accuracy of PCA.

The results of the second trial show the performance of four ML algorithms applied on all extracted measurements, as can be seen in Table 3. Table 3 shows each model’s accuracy before applying k-fold cross-validation and PCA in the third column. Accuracy after applying only k-fold cross-validation is shown in the fourth column. Accuracy after applying only PCA is shown in the fifth column. In the sixth column, the accuracy after applying k-fold cross-validation and PCA is presented. Recall score and precision for each model are shown in the seventh and eighth columns, respectively. Figure 9 shows the accuracy, recall score, and precision of ML classifiers after applying k-fold cross-validation and PCA in this trial.

## 5. Discussion

Due to no clear evidence for selection of measurements in the related works, such as in [46], which used only cortical thickness, our first trial experiments’ results show the best measurements among four measurements that are commonly used in drug-resistant epilepsy: cortical thickness, grey matter volume, white matter volume, and cerebrospinal fluid volume. The SVM model that classifies the white matter volume and the grey matter volume (Exp2) outperforms the SVM model that classifies cortical thickness (Exp1) and the SVM that classifies cerebrospinal fluid (Exp3). We can conclude that the grey and white matter volumes are the most significant measurements among the four common measurements, as the model that classified using them obtained 80.00% accuracy, 83.00% recall score, and 83.00% precision. We noticed that combining grey matter and white matter volume with the cortical thickness, as in Exp6, increased the SVM’s recall score to 83.33% and the precision to 83.33%. What seems to stand out is grey matter volume and white matter volume can stand alone to achieve the highest accuracy (80.00%), as in Exp2. Figure 7 displays the accuracy, recall score, and precision for the seven experiments in the first trial.

Our second trial relied on all eight extracted measurements: mean thickness (mm), grey matter volume (mm${}^{3}$), cerebrospinal fluid volume (mm${}^{3}$), white matter volume (mm${}^{3}$), total volume of grey matter and white matter (mm${}^{3}$), cortical area mid (mm${}^{2}$), cortical area inner (mm${}^{2}$), and cortical area pial (mm${}^{2}$). Based on our literature review, this is the first time that this has been performed in this manner. Furthermore, it examined the impact of the feature reduction technique using PCA. Additionally, it made comparisons between the performances of four different ML classifiers: LR, K-NN, SVM, and naive Bayes. This is also the first time that this has been performed based on the aforementioned works. The K-NN (n_neighbours = 5) model outperformed the other machine learning classifiers, with 97.11% accuracy, 75.00% recall score, and 75.00% precision. We noticed that PCA (n_ components = 20) increased the performance of K-NN model with n_neighbours = 5. So we can conclude that feature reduction technique is effective in this field.

## 6. Conclusions

In this study, we applied two ML trials to classify the extracted measurements from T1-weighted MRI for drug-resistant epilepsy patients and healthy control subjects. Data were collected from King Abdulaziz University hospital. Preprocessing on images was performed using BrainSuite’s pipeline. The first trial was performed on seven combinations of standard measurements: cortical thickness, white matter volume, grey matter volume, and cerebrospinal fluid volume. The white and grey matter volumes are the most significant of these standard measurements. The second trial used four different ML classifiers: LR, K-NN, SVM, and naive Bayes. Results were considered both before and after applying k-fold cross-validation and PCA. It can be seen that the k-nearest neighbours classifier outperformed the other ML classifiers. Feature reduction techniques such as PCA are successful in classification of extracted drug-resistant epilepsy measurements.

## 7. Recommendations and Future Work

Features extracted from T1-weighted MRI images are effective in drug-resistant epilepsy classification. However, different methods can be used to select the best features. Deep learning can also be considered for performing the classification.

We are looking to use DL approaches to extract and compare features with the current extracted features.

## Figures and Tables

**Figure 1 brainsci-13-00487-f001:**
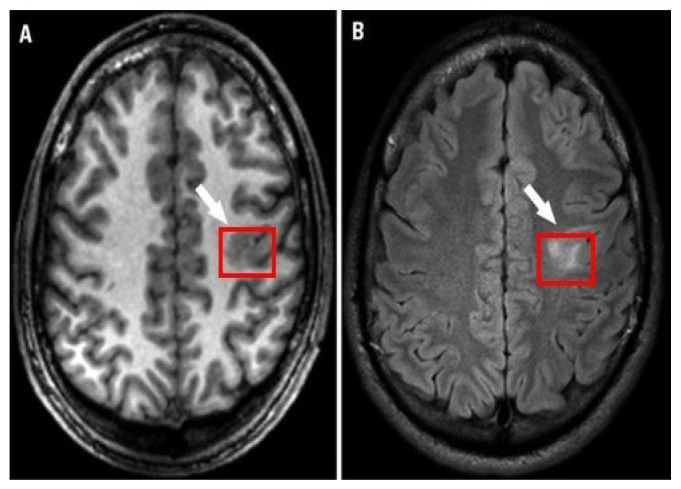
Focal cortical dysplasia lesion (**A**) T1-weighted MRI and (**B**) FLAIR MRI [10].

**Figure 2 brainsci-13-00487-f002:**
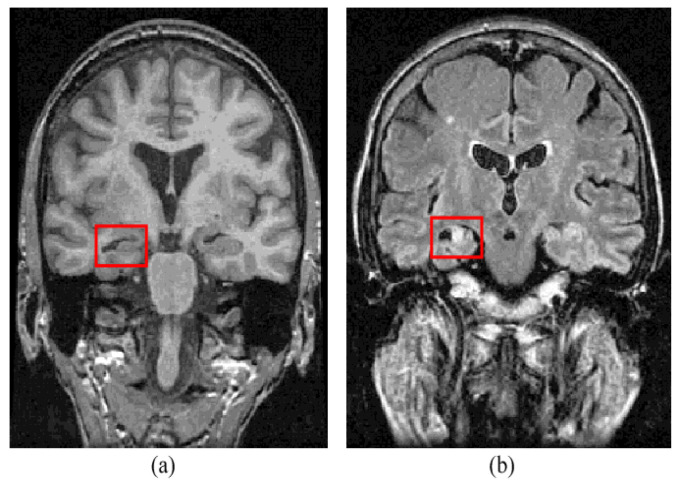
Mesial temporal sclerosis lesion (**a**) T1-weighted MRI and (**b**) FLAIR MRI [11].

**Figure 3 brainsci-13-00487-f003:**
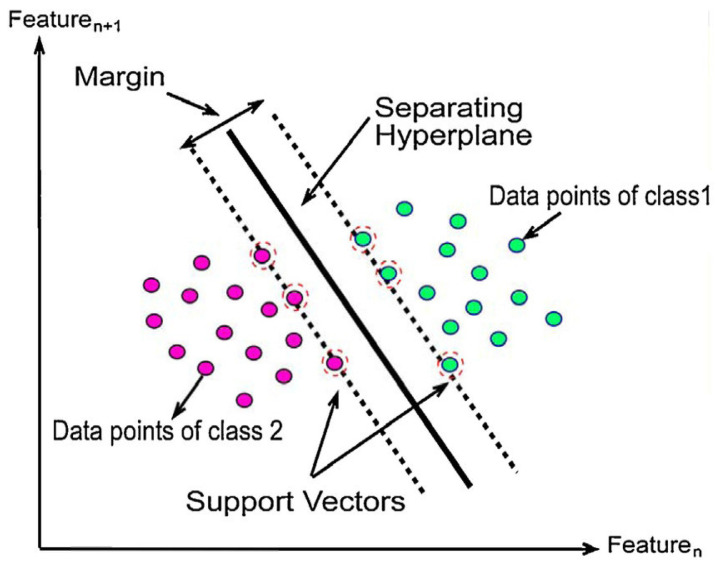
SVM hyperplane for linearly separable data points [34].

**Figure 4 brainsci-13-00487-f004:**
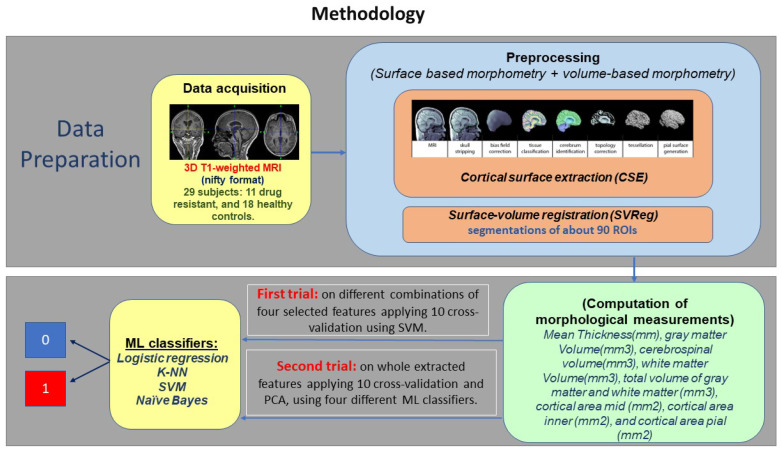
Schematic of the study.

**Figure 5 brainsci-13-00487-f005:**
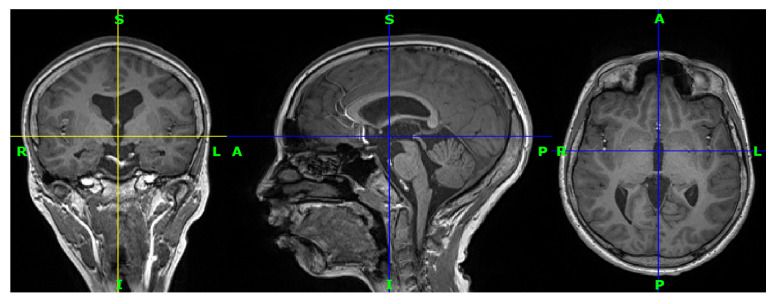
Three-dimensional T1-weighted MRI for a patient from the data set in coronal view, sagittal view, and axial view, respectively. R indicates the right of the brain, L indicates the left of the brain, S (superior) indicates the top of the body, and I (inferior) indicates the bottom of the body.

**Figure 6 brainsci-13-00487-f006:**
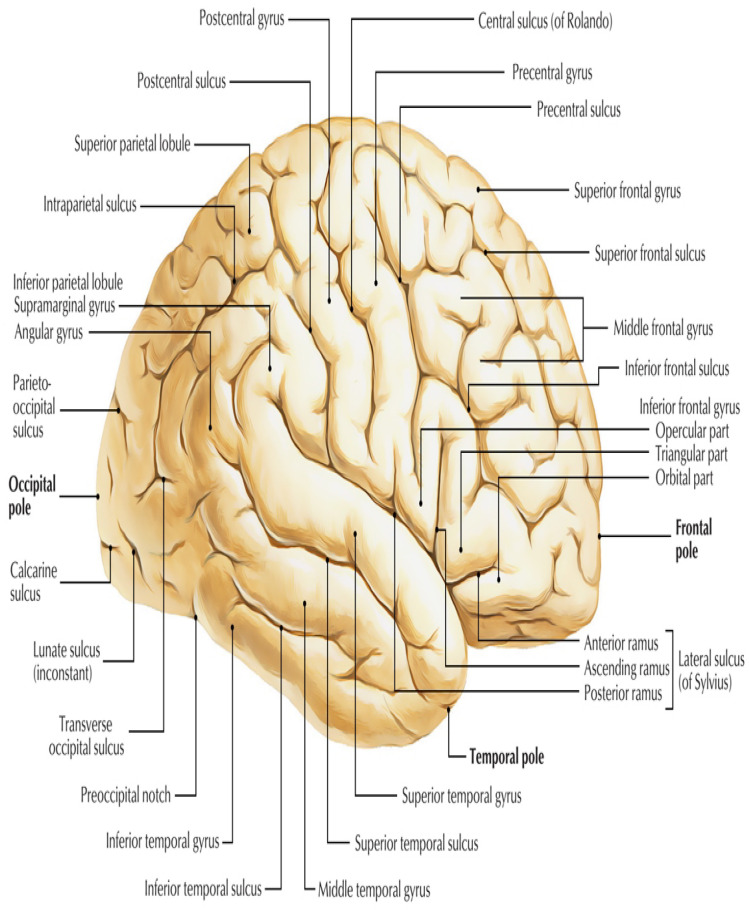
Some of the ROI of the brain [60].

**Figure 7 brainsci-13-00487-f007:**
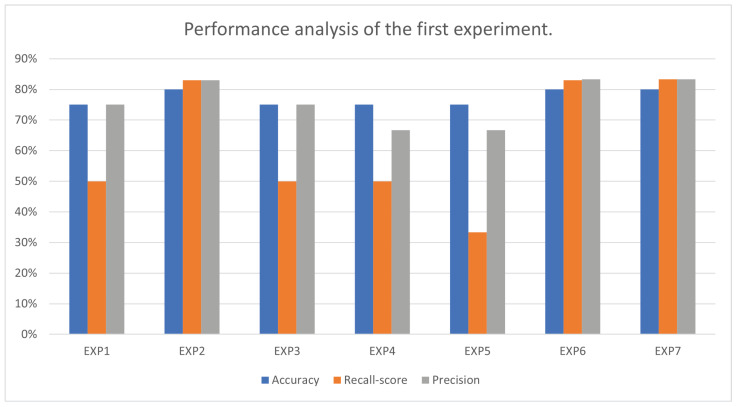
Accuracy, recall score, and precision of different combinations of measurements in the first experiment.

**Figure 8 brainsci-13-00487-f008:**
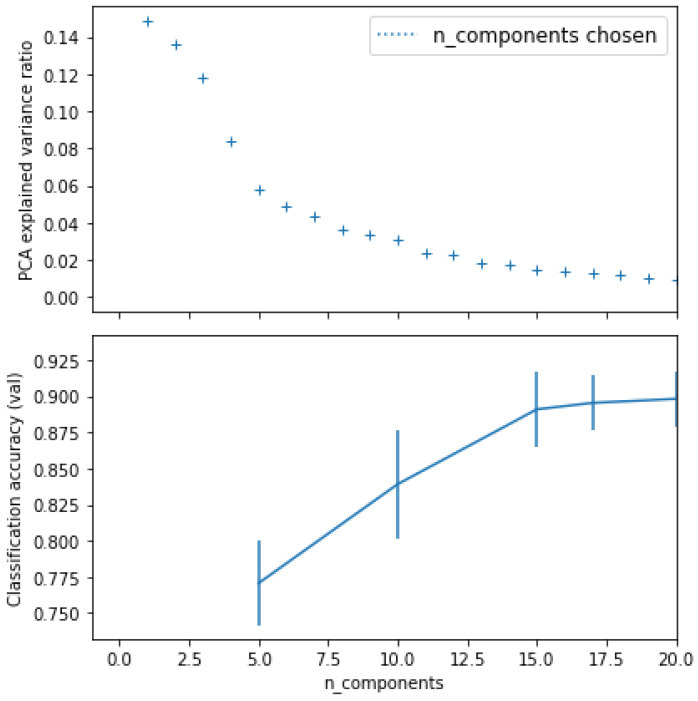
Variance ratio and classification accuracy of principal component analysis (PCA).

**Figure 9 brainsci-13-00487-f009:**
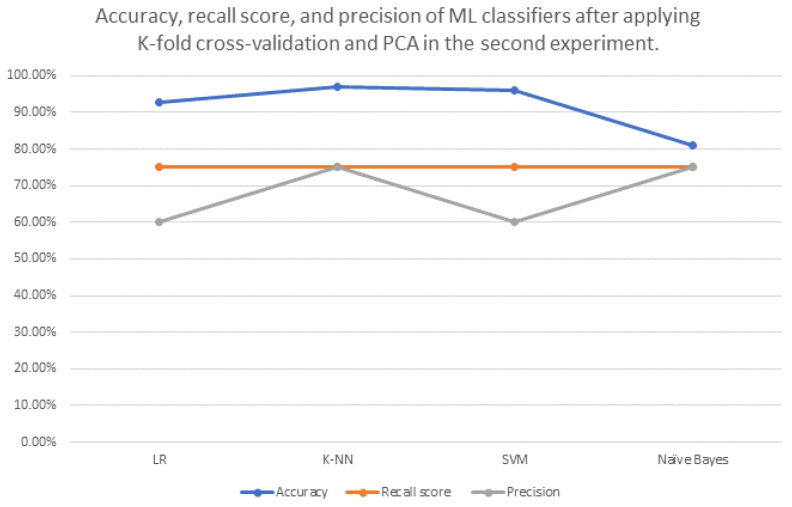
Accuracy, recall score, and precision of ML classifiers after applying K-fold cross-validation and PCA in the second trial.

**Table 1 brainsci-13-00487-t001:** The region of interest (ROI) used in the statistical calculations. R refers to the right region of the brain, and L refers to the left region.

ROI_ID	Name of Region	ROI_ID	Name of Region	ROI	Name of Region
120	R. superior frontal gyrus	226	R. angular gyrus	500	R. Insula
121	L. superior frontal gyrus	227	L. angular gyrus	501	L. Insula
130	R. middle frontal gyrus	228	R. superior parietal gyrus	612	R. caudate nucleus
131	L. middle frontal gyrus	229	L. superior parietal gyrus	613	L. caudate nucleus
142	R. pars opercularis	242	R. pre-cuneus	614	R. putamen
143	L. pars opercularis	243	L. pre-cuneus	615	L. putamen
144	R. pars triangularis	310	R. temporal pole	616	R. globus pallidus
145	L. pars triangularis	311	L. temporal pole	617	L. globus pallidus
146	R. pars orbitalis	322	R. superior temporal gyrus	620	R. nucleus accumbens
147	L. pars orbitalis	323	L. superior temporal gyrus	621	L. nucleus accumbens
150	R. pre-central gyrus	324	R. transverse temporal gyrus	630	R. claustrum
151	L. pre-central gyrus	325	L. transverse temporal gyrus	631	L. claustrum
162	R. transvers frontal gyrus	326	R. middle temporal gyrus	640	R. thalamus
163	L. transvers frontal gyrus	327	L. middle temporal gyrus	641	L. thalamus
164	R. gyrus rectus	328	R. inferior temporal gyrus	650	R. basal forebrain
165	L. gyrus rectus	329	L. inferior temporal gyrus	651	L. basal forebrain
166	R. middle orbito-frontal gyrus	330	R. fusiforme gyrus	660	R. lateral geniculate nucleus
167	L. middle orbito-frontal gyrus	331	L. fusiforme gyrus	661	L. lateral geniculate nucleus
168	R. anterior orbito-frontal gyrus	342	R. parahippocampal gyrus	662	R. medial geniculate nucleus
169	L. anterior orbito-frontal gyrus	343	L. parahippocampal gyrus	663	L. medial geniculate nucleus
170	R. posterior orbito-frontal gyrus	344	R. hippocampus	670	R. superior colliculus
171	L. posterior orbito-frontal gyrus	345	L. hippocampus	671	L. superior colliculus
172	R. lateral orbitofrontal gyrus	346	R. amygdala	680	R. inferior colliculus
173	L. lateral orbitofrontal gyrus	347	L. amygdala	681	L. inferior colliculus
182	R. paracentral lobule	422	R. superior occipital gyrus	690	R. mamillary body
183	L. paracentral lobule	423	L. superior occipital gyrus	691	L. mamillary body
184	R. cingulate gyrus	424	R. middle occipital gyrus	701	L. Ventricular System
185	L. cingulate gyrus	425	L. middle occipital gyrus	720	R. lateral ventricle
186	R. subcallosal gyrus	442	R. inferior occipital gyrus	740	third ventricle
187	L. subcallosal gyrus	443	L. inferior occipital gyrus	800	Brainstem
222	R. post-central gyrus	444	R. lingual gyrus	900	Cerebellum
223	L. post-central gyrus	445	L. lingual gyrus	2000	White matter (cerebrum)
224	R. supramarginal gyrus	446	R. cuneus		
225	L. supramarginal gyrus	447	L. cuneus		

**Table 2 brainsci-13-00487-t002:** Performance analysis of SVM classifier on different combinations of selected measurements.

Experiment No.	Accuracy	Best Parameters for Accuracy	Recall Score	Precision
EXP1	75.00%	‘C’: 0.25 ‘kernel’: ‘linear’	50.00%	75.00%
EXP2	80.00%	‘C’: 0.25 ‘kernel’: ‘linear’	83.00%	83.00%
EXP3	75.00%	‘C’: 0.25 ‘gamma’: 0.1 ‘kernel’: ‘rbf’	50.00%	75.00%
EXP4	75.00%	‘C’: 0.25 ‘kernel’: ‘linear’	50.00%	66.66%
EXP5	75.00%	‘C’: 0.25 ‘gamma’: 0.1 ‘kernel’: ‘rbf’	33.33%	66.66%
EXP6	80.00%	‘C’: 0.25 ‘kernel’: ‘linear’	83.33%	83.33%
EXP7	80.00%	‘C’: 0.25 ‘kernel’: ‘linear’	83.33%	83.33%

**Table 3 brainsci-13-00487-t003:** Performance analysis of different classifiers on all extracted measurements.

ML Classifier Name	Best Parameters for Accuracy (If Exist)	Accuracy before Applying K-Fold Cross Validation and PCA	Accuracy after Applying K-Fold Cross Validation Only	Accuracy after Applying PCA Only	Accuracy after Applying K-Fold Cross Validation and PCA	Recall Score	Precis Ion
Logistic Regression (LR)	-	66.67%	90.00%	(n_components = 20) 66.67%	92.82%	75.00%	60.00%
K-Nearest Neighbours (K-NN)	metric: ‘minkowski’, and n_neighbours = 5	77.78%	55.00%	(n_components = 20) 77.78%	97.11%	75.00%	75.00%
Support vector machine (SVM)	C = 0.25, and kernel: ‘linear’	66.67%	90.00%	(n_components = 20) 66.67%	96.05%	75.00%	60.00%
Naive Bayes “GaussianNB”	-	77.78%	70.00%	(n_components = 20) 77.78%	81.14%	75.00%	75.00%

## Data Availability

Not applicable.

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
