# Peer review of "Detecting Cortical Thickness Changes in Epileptogenic Lesions Using Machine Learning"

_brainsci, 2023, doi:10.3390/brainsci13030487_

Round 1

Reviewer 1 Report

The paper's subject is interesting as diagnosing neurological disorders is gaining much attention from researchers. The authors used a dataset provided by King Abdulaziz Hospital, which adds value to the study. 

The Literature review covered the most important works related to the subject and the research methodology; the assessment criteria were well presented. 

In the first experiment, the authors applied the SVM classifier to different combinations of the relevant features. The best parameters of the SVM were selected using GridSearchCV. The obtained results were promising.

The second experiment, including the feature selection step, helped improve the classification accuracy to 90%. The obtained results are auspicious. 

It would be interesting to add some information regarding the state-of-the-art results in the discussion section. This can be easy as many results are mentioned in the literature review. 

writing errors: 

in 283 a reference is missing

in 312 a reference is missing 

Reviewer 2 Report

The authors use machine learning technology to detect the pathological changes of epilepsy, which has important research significance. However, the content of the article is too simple, especially the discussion and conclusion, which makes the whole paper more like an experimental record. In addition, the format of the paper does not conform to the regulations of the journal, and there are many mistakes in the way of citation.

Reviewer 3 Report

This work is well within the scope of Brain Sciences and it may be of interest to most of the readers of this journal. It was clear that the main contribution of this work was to investigate if we can use deep learning to perform and select the best features from T1-weighted MRI images extracted features that are efficient in drug-resistant epilepsy classification.

The manuscript shows introductory background material sufficient for someone, not an expert in this area to understand the context and significance of this work, with good references to follow.

 This research has reached beneficial conclusions that could be summarized as saying that machine learning is very promising in this area as well, but the steps so far have specific drawbacks that make it impossible to generalize and propose a specific protocol that could be widely adopted in the specific scientific field. Please could you explain to me in detail if the conclusions of machine learning were applied and what were the results of this application of machine learning in relation to not using it? Otherwise, the conclusion reached by the research team could serve as a reference guide, but with the aim of verifying it from medical data, so that it can be generalized.

A serious weakness of this research is the fact that Turnitin returned a similarity index of 68% including the references, it is identified by an almost identical student work at King Abdulaziz University and 2% with the University of Stellenbosch, so as you understand this work cannot be accepted in this form.

Please, since the conclusions are really very worthwhile rewrite the corresponding points of the manuscript and I would be happy to see and review the final text.

In conclusion, this manuscript is NOT OK, according to Turnitin shows high plagiarism. The English in this paper is good, except for some grammatical mistakes across the text that need proofreading.

Finally, for all the above and the specific comment below, I have opted to recommend Reconsider After Major Revision.

Round 2

Reviewer 3 Report

Dear Authors, thank you so much for taking the effort to respond to the comments I made.

I have all the will to accept the justification you gave me and to exclude the specific scientific forum, but in case it is also published in the journal, obviously, the originality of your new work is lost. The contribution to the particular scientific journal is indeed yours, but it is detected by the Turnitin software as published research.

If it is not a publication then what is it, please?

A serious weakness of this new research is the fact that Turnitin returned a similarity index of 81% excluding the quotes and the bibliography.

In conclusion, this manuscript is NOT OK, according to Turnitin shows high plagiarism.

I am very sorry, but in order to accept the publication the text will have to be changed so that it is genuinely original.